# APOE ε4 in Depression-Associated Memory Impairment—Evidence from Genetic and MicroRNA Analyses

**DOI:** 10.3390/biomedicines10071560

**Published:** 2022-06-30

**Authors:** Sarah Bonk, Kevin Kirchner, Sabine Ameling, Linda Garvert, Henry Völzke, Matthias Nauck, Uwe Völker, Hans J. Grabe, Sandra Van der Auwera

**Affiliations:** 1Department of Psychiatry and Psychotherapy, University Medicine Greifswald, 17489 Greifswald, Germany; sarah.bonk@med.uni-greifswald.de (S.B.); kevin.kirchner@med.uni-greifswald.de (K.K.); linda.garvert@uni-greifswald.de (L.G.); hans.grabe@med.uni-greifswald.de (H.J.G.); 2Interfaculty Institute for Genetics and Functional Genomics, University Medicine Greifswald, 17475 Greifswald, Germany; sabine.ameling@uni-greifswald.de (S.A.); uwe.voelker@uni-greifswald.de (U.V.); 3German Centre for Cardiovascular Research (DZHK), Partner Site Greifswald, 17475 Greifswald, Germany; voelzke@uni-greifswald.de (H.V.); matthias.nauck@med.uni-greifswald.de (M.N.); 4Institute for Community Medicine, University Medicine Greifswald, 17475 Greifswald, Germany; 5Institute of Clinical Chemistry and Laboratory Medicine, University Medicine Greifswald, 17475 Greifswald, Germany; 6German Centre for Neurodegenerative Diseases (DZNE), Site Rostock/Greifswald, 17489 Greifswald, Germany

**Keywords:** amyloid beta, Alzheimer’s disease, APOE ε4, MDD, memory, miRNA, depression, cognition

## Abstract

(1) Background: The aim of this study was to replicate a reported interaction between APOE ε4 status and depression on memory function in two independent, nondemented samples from the general population and to examine the potential role of circulating plasma miRNAs. (2) Methods: The impact of the APOE ε4 allele on verbal memory and the interaction with depression is investigated in two large general-population cohorts from the Study of Health in Pomerania (SHIP, total *n* = 6286). Additionally, biological insights are gained by examining the potential role of circulating plasma miRNAs as potential epigenetic regulators. Analyses are performed using linear regression models adjusted for relevant biological and environmental covariates. (3) Results: Current depression as well as carrying the APOE ε4 allele were associated with impaired memory performance, with increasing effect for subjects with both risk factors. In a subcohort with available miRNA data subjects with current depressive symptoms and carrying APOE e4 revealed reduced levels of hsa-miR-107, a prominent risk marker for early Alzheimer’s Disease. (4) Conclusions: Our results confirm the effect of depressive symptoms and APOE ε4 status on memory performance. Additionally, miRNA analysis identified hsa-miR-107 as a possible biological link between APOE ε4, depressive symptoms, and cognitive impairment.

## 1. Introduction

Major depressive disorder (MDD) is one of the most common mental disorders worldwide with an estimated lifetime prevalence of about 20.6% in the United States [1]. It is characterized by symptoms such as decreased mood, anhedonia, feelings of guilt, lack of concentration, low self-esteem, disability, increased mortality, or increased risk for suicide [2]. Furthermore, core features of MDD are deficits in executive functioning, processing speed, as well as impairments in learning and memory [3], especially in patients with recurrent episodes [4]. This is supported by magnetic resonance imaging (MRI) and functional MRI (fMRI) analyses, which suggest atrophies in the hippocampus, amygdala, and certain areas of the prefrontal cortex (PFC), as well as a decreased connectivity between these areas [5].

Previous studies on Alzheimer’s disease (AD) showed that amyloid β (Aß) deposits and atrophies in the hippocampus are associated with memory impairments but also with apolipoprotein E (APOE) ε4 status [6]. APOE ε4 is one of three isoforms (ε2, ε3, and ε4) of the human APOE gene, which is located on chromosome 19 and whose encoding proteins merely differ in two amino acids [6,7]. The ε4 allele is the major genetic risk factor for AD and associated with memory impairments [8]. Associations have also been reported for other neuropsychiatric conditions, such as epilepsy, MDD, or schizophrenia [9,10]. As memory impairment represents an early symptom of AD, researchers are increasingly interested in the link between memory impairment and the APOE ε4 function. In a large genome-wide association study (GWAS), the CHARGE consortium could link the APOE locus to verbal declarative memory especially in older subjects [11]. However, this association remains controversial due to inconsistent findings [6].

Given the similarity of cognitive symptoms in MDD and AD, researchers suspect a link between both disorders. As the risk for MDD and AD increases with age and both diseases co-occur in varying time-sequences, both may also be connected by common biological mechanisms. Likewise, early depressive episodes may increase the risk for later AD, and on the other hand, AD patients show increased vulnerability to developing a depressive episode [12].

As MDD and APOE ε4 status are both associated with cognitive impairment, Piers and colleagues [13] also assumed that the APOE ε4 allele might influence the association between depressive symptoms and cognitive impairment. Their results showed that depressive but nondemented individuals exhibited poorer short- and long-term visual memory performance when additionally carrying the APOE ε4 allele.

In addition to genetic variation, epigenetic mechanisms such as microRNAs (miRNAs) have been investigated as biomarkers for complex traits. MiRNAs are small (≈22 nt) noncoding RNAs that play important regulatory roles by targeting messenger RNAs (mRNAs) for degradation, mediating translational repression, or, in rare cases, enhancing transcription by binding to the promoter region of target genes [14,15]. Gene regulation by miRNAs is far-reaching as one miRNA can interact with many hundreds of mRNAs and one mRNA can be targeted by several miRNAs. Previous studies could identify certain miRNAs as biomarkers in association with depression, AD, or both traits [16,17,18].

Following these previous findings, the first aim of our study was to replicate the results of Piers and colleagues [13] on the interaction between APOE ε4 status and depression on memory function in two independent, nondemented samples from the general population. In extension to Piers and colleagues [13], we specifically distinguished between current depressive symptoms and lifetime MDD status. Secondly, we additionally examined the potential role of circulating plasma miRNAs to gain first insights into epigenetic regulators of depression, the APOE ε4 allele, and memory impairment.

## 2. Materials and Methods

### 2.1. Study Population

We analyzed data from the Study of Health in Pomerania (SHIP) [19] comprising adult German residents in north-eastern Germany. A number of 4308 Caucasian subjects participated at baseline (SHIP-START-0; 1997–2001). To date, three regular follow-ups have been carried out (SHIP-START-1 2002–2006; START-2 2008–2012; START-3 2014–2016). In parallel to SHIP-START-2, detailed assessments of life events and mental disorders in these subjects were conducted within SHIP-LEGEND (Life Events and Gene–Environment Interaction in Depression, hereafter referred to as LEGEND). From 2007 to 2010, 2400 participants from the baseline SHIP-START-0 sample participated in this study [20]. Thus, SHIP-START-2 and SHIP-LEGEND are two follow-up waves of the same baseline cohort SHIP-START-0, and 1944 subjects from SHIP-START-0 participated in both follow-ups, SHIP-START-2 and SHIP-LEGEND. From 2008 to 2012, a new independent sample called SHIP-TREND-0 (*n* = 4420; hereafter referred to as TREND) was drawn from the same area, encompassing similar examinations as in SHIP-START-2. All participants underwent a standardized computer-assisted personal interview, during which they provided information on sociodemographic and lifestyle factors.

The investigations in both studies were carried out in accordance with the Declaration of Helsinki, including written informed consent of all participants. Survey and study methods were approved by the institutional review boards of the University of Greifswald.

### 2.2. Penotype Measures

#### 2.2.1. Verbal Memory

In TREND, the word list of the Nuremberg Age Inventory (NAI) was used as a measure for immediate- and delayed memory performance. The NAI is a German test developed to measure cognitive abilities during brain aging [21,22]. The number of correctly identified words is summarized to a sum score minus the number of wrongly identified distractor words for short- and long-term retrieval, respectively.

Subjects in LEGEND were administered the auditory Verbal Learning and Memory Test (VLMT), a German adaption of the widely used Rey Auditory Verbal Learning Test [23]. The VLMT was used to assess short-term learning as well as delayed retrieval [22]. The number of correctly remembered words for short- and long-term retrieval was stored in two separate sum scores.

#### 2.2.2. Depression

In TREND and LEGEND, a diagnostic interview for mental disorders was performed based on the fourth edition of the Diagnostic and Statistical Manual for Mental Disorders (DSM-IV) [24,25], which also includes the diagnosis of major depressive disorder (MDD). Current depressive symptoms were assessed in TREND using the patient health questionnaire (PHQ-9) and in LEGEND using the Beck depression inventory (BDI-II). The PHQ-9 is a 9-item self-report questionnaire also with high reliability and validity [26]. The BDI-II measures current depressive symptoms with high reliability and validity using a 21-item self-report questionnaire [27]. For both scores, higher values reflect a higher load of depressive symptoms. In both samples, general mental health was measured using the mental health composite score (MCS) of the Short Form-12 Health Survey (SF-12) [28] using the original scoring method by Ware and colleagues [29]. For the MCS, lower values reflect a poorer overall mental health. Although the MCS does not measure depressive symptoms directly, it is a broad measure of current mental status and could enable a higher comparability between the cohorts as it was available in TREND and LEGEND.

#### 2.2.3. Covariates

In TREND, education measured as the number of schooling years was divided into three categories according to the German school system: less than 10 years, exactly 10 years, and more than 10 years. Smoking status was divided into never, former-, and current smokers. Heart diseases included the self-report of a diagnosis of angina pectoris, heart attack, stroke, atrial fibrillation, or heart insufficiency. Hypertension included an increased systolic (≥140 mmHg) or diastolic blood pressure (≥90 mmHg) or a self-reported antihypertensive medication during the last year. Alcohol consumption was measured using the question “How often have you drunk an alcoholic beverage within the last 12 months?” and is categorized into never (1), once per month or less (2), two to four times per month (3), two to three times per week (4), and more than three times per week (5). Diabetes was diagnosed if a subject fulfilled at least one out of four criteria: self-report of the disease, intake of diabetes medication, Hba1c levels ≥ 6.5%, or serum glucose level ≥ 11.1 mmol/L.

In LEGEND, these covariates were not available, as this was a purely psychiatric study. As SHIP-START-2 was conducted in parallel to LEGEND, the data for education, hypertension, smoking, diabetes, and heart diseases were taken from SHIP-START-2. Education, hypertension, and smoking were defined identically as in TREND. Diabetes was calculated similar to TREND, but the diagnosis via self-report now included the information from SHIP-START-0, SHIP-START-1, or SHIP-START-2. Heart diseases included the self-report of a diagnosis of angina pectoris, heart attack, stroke, atrial fibrillation, or heart insufficiency in SHIP-START-0 or SHIP-START-1. To take the time interval between LEGEND and SHIP-START-2 into account, the time between both studies (in years) was used as a further covariate.

### 2.3. Omics Data

#### 2.3.1. Genome-Wide SNP Chip

Single-nucleotide polymorphism (SNP) information was taken from the genetic data in SHIP. The baseline SHIP-START sample (*n* = 4070) was genotyped using the Affymetrix Genome-Wide Human SNP Array 6.0. Genotyping of a subset of the SHIP-TREND subjects (*n* = 986) was performed using the Illumina Infinium HumanOmni 2.5 Bead Chip. The remaining SHIP-TREND sample (*n* = 3134) was genotyped using the Illumina Infinium GSA. Imputation of genotypes was performed using the HRCv1.1 reference panel and the Eagle and minimac3 software implemented in the Michigan Imputation Server for pre-phasing and imputation, respectively. For more details, see Völzke and colleagues [19]. SNPs with a Hardy–Weinberg-Equilibrium *p*-value < 0.0001, a call rate < 0.95, and a MAF < 1% were removed before imputation.

#### 2.3.2. APOE Status

The APOE genotypes were determined on the basis of the two single-nucleotide polymorphisms rs429358 (C; T) and rs7412 (T; C) from the resulting imputation (imputation quality >0.8; Hardy–Weinberg Equilibrium, *p* > 0.05) [30,31]. As we used the data from the genome-wide SNP chip instead of strand-specific genotyped SNPs for determination of APOE status, two ambiguous SNP combinations occurred where APOE ε2/ε4 and ε1/ε3 could not be discriminated (http://www.snpedia.com/index.php/APOE; accessed on 15 May 2022). Those participants were excluded from the genetic analyses (SHIP-START *n* = 67; SHIP-TREND *n* = 99). Subjects were defined as APOE ε4 carriers if they had at least one ε4-allele. 

#### 2.3.3. miRNA Measurement

MiRNA data were available in a subsample of TREND. Details on profiling and quality control are provided in the supplement or elsewhere [32,33]. The miRNAs used in this analysis were measured in two different batches (batch 1: *n* = 371; batch 2: *n* = 337). In the preprocessing step of RT-qPCR (real time quantitative PCR) data, a miRNA was selected for further analysis if at least, in 40% of the samples, a Cycle-threshold (Ct) ≤ 37 was detected for each batch separately. In order to consider the influence of technical parameters, the Ct values of synthetic spiked-in miRNAs—monitoring the efficiency of miRNA extraction (UniSp2 and the difference between Ct values of UniSp4 and UniSp2)—the interplate calibrator (UniSp3), as well as the age of the biosamples (dt_biobank) were regressed out of the data. Linear regression was performed for each miRNA according to the model ∆Ct ~ UniSp2 + UniSp4-UniSp2 + UniSp3 + dt_biobank, which treats the ∆Ct values as dependent variables and the technical parameters as independent variables. The resulting residuals were used as dependent variables in later models to detect associations between miRNAs and phenotypes of interest.

### 2.4. Statistical Analyses

#### 2.4.1. Main Analyses in TREND

Direct Effects on NAI in TREND

Ordinary least squares linear regression models were applied to investigate the association of current depressive symptoms (PHQ-9 and MCS) and lifetime depression (MDD) with the verbal memory scores (NAI; immediate and delayed recall) in TREND.

Ordinary least squares linear regression models were applied to investigate the association of the APOE ε4 allele with the verbal memory scores in TREND.

Interactions between Depression and APOE ε4 Allele on NAI in TREND

Ordinary least squares linear regression models were applied to investigate the interaction between current depressive symptoms (PHQ-9 or MCS), or lifetime depression (MDD) and APOE ε4 status on verbal memory scores in TREND. 

All regression models in TREND were adjusted for two different sets of covariates. The first set included age, sex, and education. The second adjustment set additionally included hypertension, smoking status, alcohol intake, diabetes, and heart disease for better comparison with Piers and colleagues [13]. All analyses including the APOE ε4 allele were also adjusted for the first three genetic principal components and genetic batch. A *p*-value of *p* < 0.10 was set as the significance level as this was a replication analysis of a previous finding by Piers and colleagues [13].

#### 2.4.2. Replication in LEGEND

Ordinary least squares linear regressions for direct effects (BDI-II, MCS, MDD, and APOE ε4) and interaction effects on verbal memory scores (VLMT) were applied. As LEGEND was a psychiatric study, information on medical variables was not available. However, these variables were part of the SHIP-START-2 examination. Unfortunately, merging data from SHIP-START-2 and LEGEND was not trivial. One reason is that not all subjects from LEGEND participated in SHIP-START-2, which would reduce the sample size. Furthermore, the mean time difference between both examinations was 1.6 years (−1.8 up to 4.9 years) for those that participated in both studies, which is a long time especially for variables based on blood measurements. As both parameters, verbal memory outcome and current depressive symptoms, are not stable traits, sensitivity analysis in LEGEND should be treated with caution with a focus on the results for covariate set 1 (age, sex, and education) and covariate set 2 as sensitivity analysis (age, sex, education, hypertension, smoking status, diabetes, heart disease, and time between SHIP-START-2 and LEGEND). All analyses including APOE ε4 status were also adjusted for the first three genetic principal components. A *p*-value of *p* < 0.10 was set as the significance level for the replication.

#### 2.4.3. Sensitivity Analyses for miRNAs in TREND Subsample

Ordinary least squares linear regression analyses were performed to assess the predictive value of the direct effects, as well as the multiplicative interaction terms between depression and APOE ε4 status on miRNA residuals. Additionally, direct effects of miRNA levels on verbal memory were investigated. Analyses were adjusted for the miRNA batch, three genetic principal components, blood cell parameters (hematocrit (HCT) and platelet count), several environmental factors (smoking status, education, and body mass index (BMI)), as well as sex and age (nonlinear as splines). A batch was included in the analysis for a specific miRNA if at least 100 subjects contained a valid measurement of the respective miRNA (see supplement for a complete list of miRNAs used).

All reported *p*-values for the miRNA analyses were two-sided. Multiple testing correction was applied using the Benjamini–Hochberg correction.

All reported analyses were performed with R version 3.6.3 [34].

## 3. Results

The characteristics of both cohorts are given in Table 1. Significant differences between the subjects in TREND and LEGEND were apparent for age (*p* < 2.2 × 10^−16^), education (*p* = 1.64 × 10^−10^), MCS (*p* = 0.007), smoking (*p* = 9.9 × 10^−8^), diabetes (*p* = 0.001), hypertension (*p* = 3.8 × 10^−9^), and heart disease (*p* < 2.2 × 10^−16^). Higher rates for diseases might be due to an overall higher age of the cohort. The age differences were caused by the different survey waves of LEGEND (follow-up of SHIP-START) and TREND (new baseline cohort). The differences in education were caused by the different time periods of the studies (SHIP-START 1997–2001 and SHIP-TREND 2008–2012). The correlation between the two scores for depressive symptoms was r = −0.62 in TREND (PHQ-9 and MCS) and r = −0.70 in LEGEND (BDI-II and MCS).

### 3.1. Direct Effects of Depression and APOE ε4 Status on Verbal Memory in TREND

We found direct effects of depressive symptoms (PHQ-9 and MCS) on NAI (immediate and delayed recall of words) for both covariate sets in TREND. A higher load of depressive symptoms (higher scores for PHQ-9; lower scores for MCS) was associated with lower scores for verbal memory. No direct effect of lifetime depression on verbal memory was found in TREND. The effect of APOE ε4 status on verbal memory was significant for the immediate and delayed recall of words for both covariate sets. APOE ε4 carriers revealed overall decreased verbal memory scores. All *p*- and β-values for the effects of depression variables or APOE ε4 status on verbal memory (immediate and delayed recall) are summarized in Table 2.

### 3.2. Interaction Effects between Depression and APOE ε4 Status on Verbal Memory in TREND

We found a significant interaction effect between current depressive symptoms and APOE ε4 status on immediate recall, but not on delayed recall of words. For PHQ-9, this effect could only be observed for covariate set 2 (*p* = 0.083), whereas, for MCS, the interaction was significant for both covariate sets (set 1 *p* = 0.097; set 2 *p* = 0.087). No significant interaction effect between MDD and APOE ε4 status was found on NAI. The effect sizes and *p*-values for the interaction term are summarized in Table 3. A visualization for the significant interaction effects on immediate recall is given in Figure 1.

### 3.3. Replication of the Direct Effects and Interactions in LEGEND

The results for LEGEND are summarized in Table 4 (covariate set 1) and Table 5 (covariate set 2). The significant direct effects of depressive symptoms on immediate and delayed recall were replicated in both covariate sets. Additionally, the interaction term between depressive symptoms measured by BDI-II and APOE ε4 status significantly affected the immediate recall score in LEGEND in covariate set 1, while this interaction term was not significant in covariate set 2. There was no interaction effect of MCS and APOE ε4 status in both covariate sets. No significant direct effect of MDD or APOE ε4 status or their interaction on verbal memory was observed in both covariate sets.

### 3.4. miRNA Analyses in the TREND Subsample

Appendix A shows the characteristics of the miRNA batches 1 and 2. The subjects of both batches showed no remarkable differences regarding the variables of interest. There was only a significant difference for hematocrit values, which were higher for subjects in batch 2.

The final analyses included *n* = 171 miRNAs that passed the required samples sizes. For the interaction between PHQ-9 score and APOE ε4 status, three miRNAs revealed a remarkable *p*-value < 0.01 (hsa-miR-221-3p, hsa-miR-376a-3p, and hsa-let-7d-3p). None of these miRNAs survived multiple testing correction. All three miRNAs are expressed in the brain according to the miRNA tissue atlas (https://ccb-web.cs.uni-saarland.de/tissueatlas2/; accessed on 15 May 2022) [35]. 

For the interaction between MCS score and APOE ε4 status, five miRNAs showed a *p*-value < 0.01 (hsa-miR-107, hsa-miR-382-5p, hsa-miR-181a-5p, hsa-miR-99a-5p, and hsa-miR-222-3p). Again, all miRNAs are expressed in the brain. Out of this set, hsa-miR-107 survived multiple testing correction (*p* = 0.00015; pBH = 0.026) and was measured in both batches (batch 1: *n* = 338; batch 2: *n* = 315). Subjects with a poorer MCS and additionally being APOE ε4 carriers showed significantly reduced levels of hsa-miR-107 (resulting in higher Ct-values, Figure 2).

None of the miRNAs were directly associated with verbal memory score, APOE ε4 status, or depression (all *p*-values > 0.01). Results on all miRNAs with a *p*-value < 0.01 in one of the analyses can be found in Table 6.

## 4. Discussion

In summary, our results confirm the findings of Piers and colleagues [13] on the effect of depressive symptoms and APOE ε4 status on memory performance. In the nondemented general-population TREND sample, verbal short-term memory was less pronounced in subjects with acute depressive symptoms, as well as in carriers of the APOE ε4 allele. Moreover, both conditions together led to an amplification of this effect. These results could largely be replicated in our second sample LEGEND.

To explain these findings, we investigated possible connections between depression, APOE ε4 status, and cognitive impairment. Previous studies suggest that amyloid beta might be a common factor in the etiology of cognitive impairments as well as depression.

Aβ plaques have been discussed primarily in the context of AD and related cognitive deficits. They represent protease-resistant aggregates of Aβ peptides inside and outside of neurons and can be differentiated into soluble and insoluble monomers and oligomers [36]. However, instead of the larger insoluble Aβ plaques, the soluble Aβ oligomers seem to be mainly responsible for cognitive deficits [37]. These Aβ oligomers appear to exert a toxic influence on synaptic transmission by affecting synaptic long-term potentiation (LTP), as well as long-term depression (LTD), and might promote synapse loss in the hippocampus [37,38,39]. Synaptic plasticity is thought to be also impaired in depressed patients [40]. So far, increased Aβ oligomer levels were especially observed for APOE ε4 allele carriers and have been associated with an accelerated memory impairment, as well as early cognitive decline [8].

In addition to cognitive impairments, Aβ oligomers also appear in association with depressive symptoms. As a recent study showed, an intracerebroventricular injection of soluble Aβ oligomers into rat brains led to memory impairment, as well as depressive-like behavior [41]. Another study by Holderbach and colleagues [40] showed that synaptic plasticity was also impaired in an animal model of depression. Although the authors found no effect for LTP, they observed a facilitated induction of LTD. Additionally, their results suggest that anti-depressant drugs, such as selective serotonin reuptake inhibitors (SSRIs), facilitated LTP in nonstressed as well as stressed rats [40]. In addition to this, SSRIs were able to reduce Aβ42 (a cleavage variant of amyloid-precursor-protein (APP)) due to an increase in the enzymatic activity of α-secretase, which prevents the breakdown of APP [42]. Thus, SSRIs appear to have a positive effect on synaptic plasticity, as well as Aβ amyloid beta, which might explain some of their antidepressant and cognition-promoting properties.

Based on these results, we therefore suggest the following mechanism:

In the acute phase of depression, brain regeneration and synaptic plasticity are compromised due to various reasons (e.g., dysregulated HPA-axis [43] and sleeping problems with reduced glymphatic brain clearance [44]). APOE ε4 allele carriers are additionally burdened by increased Aβ oligomer levels [7,8], which trigger depressive-like behavior as well as cognitive impairment [41] due to an influence on synaptic plasticity [39].

This model is also supported by the observed difference between acute depressive symptoms and lifetime depressive disorder. If Aβ oligomers or acute depression-induced changes in neurotransmitter systems were mainly responsible for cognitive impairments, only currently depressed subjects would show these impairments in their synaptic plasticity. In addition, our considerations support the idea of Berger and colleagues [12] of neurogenesis as a possible link between depression and AD.

Regarding the observed differences in short- and long-term memory, our results are also consistent with the finding that short-term memory impairments represent an early symptom of AD [45], as this was the only memory domain affected in our analyses. Nevertheless, the underlying reason for the observed differences between verbal short- and long-term memory remains unclear. Despite the widespread view to consider short-term memory as activated long-term memory, cognition psychologists proposed them as two functionally and neurobiologically distinct systems [46]. As our results suggest, a different physiological processing mechanism might indeed underlie the short-term and long-term storage of verbal information in the hippocampus.

In recent years, epigenetic factors such as circulating miRNAs have been discussed as easily accessible biomarkers in the diagnostic process of AD and depression [47,48]. In our analysis, we identified hsa-miR-107 as a possible biological link between APOE ε4, depressive symptoms, and cognitive impairment. This miRNA has been identified in association with AD, cognition, and depression in the past [49] and is highly expressed in brain tissues [35]. The expression of hsa-miR-107 was found to be decreased [49,50] very early in AD, which is in line with observations regarding deficits in short-term verbal memory as an early symptom of AD. Although we found no direct effect of hsa-miR-107 on verbal memory, we were able to identify a biological signal for the interaction between APOE ε4 status and depression that shows a remarkable connection to the Aβ system and the progression of AD. One of the main targets of hsa-miR-107 is the β-site amyloid precursor protein-cleaving enzyme 1 (BACE1 = β-secretase 1), a gene involved in the formation of Aβ plaques [50,51]. In a first step, BACE1 promotes oligomerization of Aβ by cleaving APP [52], and the remaining membrane-bound APP fragment is then further cleaved by the gamma-secretase releasing the Aβ40/42 fragment, which forms the basis for the amyloid plaques associated with AD. As miRNAs predominantly promote degradation, destabilization, and repression of mRNA [53], a reduction of hsa-miR-107 might lead to an increased expression of BACE1, thereby initiating the cascade that increases the risk for cognitive deficits. This increased oligomerization, in turn, promotes limitations in LTP and is consistent with the findings of Wang and colleagues [41]. Moreover, this observation supports the idea of Aβ as a biological trigger of cognitive deficits in MDD and AD.

Although we were able to derive a possible biological mechanism linking APOE ε4, depressive symptoms, and cognitive impairment, several limitations of our analysis need to be acknowledged. Both samples were collected from the same study population in northeast Germany. Unfortunately, questionnaires to measure current depressive symptoms and verbal memory were not identical. This can either be viewed as a limitation or as a strength of the robustness of the results as the significant effect was generalizable. As this is still an analysis based on observational data, the hypothesized biological mechanisms need to be investigated in clinical samples and animal models of AD and depression in detail. There was also no independent sample available to replicate our miRNA results. In this general-population sample, we cannot rule out the possibility that some subjects with prodromal AD might have influenced our results. However, adjusting for current intake of antidementives or excluding these subjects did not change the significance of the results. Finally, we were only able to examine one aspect of cognition, namely verbal memory, as this was the only cognition measurement available in SHIP. This gives only a brief glimpse of the broad spectrum of cognition. Although Piers and colleagues (2021) investigated this effect in the cognitive domain of visual memory, our replication for verbal memory strengthens the plausibility of a true biological mechanism on cognition in general. Nevertheless, more research is needed to validate the biological mechanisms of depression associated cognitive impairment in association with the Aβ system.

A step in this direction has been made by Thalamuthu and colleagues [54] who investigated the interaction between MDD and genetic variants on different domains of cognition in a GWAS approach. Although their measurements of current MDD and cognition were very heterogeneous and they could not measure APOE ε4 status directly, they were able to identify the amyloid precursor protein network as an important mechanism for executive function in a considerably large sample. A genetic link between the APOE locus and verbal declarative memory has been found in a large study by the CHARGE consortium [11]; rs6857 and rs4420638). Taking their identified GWAS hits instead of the combined APOE ε4 status in our analyses results was largely comparable (data not shown), suggesting that both variables capture a similar biological signal. Thus, we recommend the use of these SNPs as proxy variables in case APOE ε4 status is not available.

Finally, our results could strengthen the findings from Piers and colleagues [13] and demonstrate the importance of more advanced research on memory impairment in the light of depression and Aβ-associated neurogenesis.

## Figures and Tables

**Figure 1 biomedicines-10-01560-f001:**
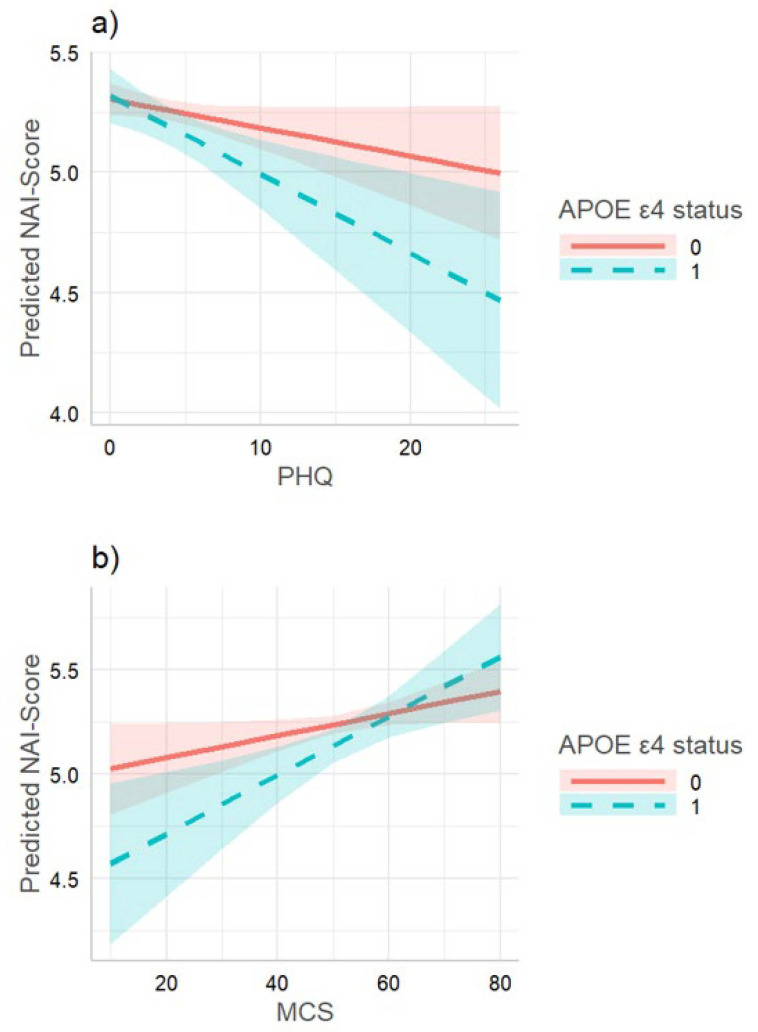
(**a**) Interaction between APOE ε4 status and PHQ-9 on immediate recall of words in TREND (covariate set 2). Subjects with high PHQ-9 scores and additionally carrying APOE ε4 revealed lower verbal memory scores than their genetic counterparts did. (**b**) Interaction between APOE ε4 status and MCS on immediate recall of words in TREND (covariate set 2). Subjects with low MCS scores and additionally carrying APOE ε4 revealed lower verbal memory scores than their genetic counterparts did.

**Figure 2 biomedicines-10-01560-f002:**
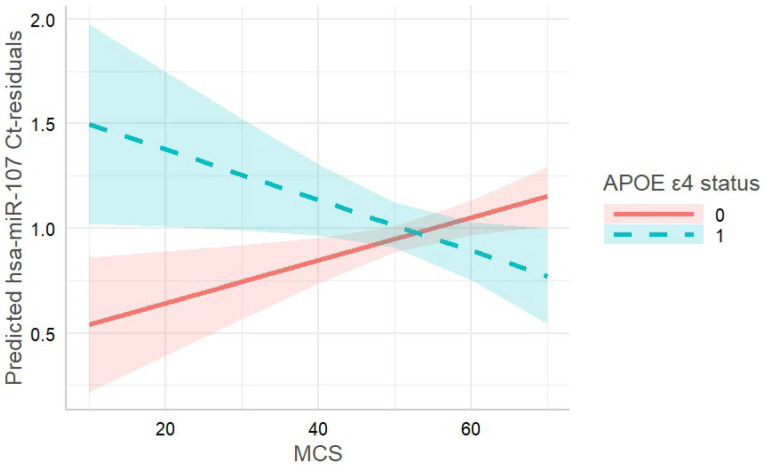
Interaction between APOE ε4 status and MCS on Ct-values for has-miR-107 in TREND subsample (*n* = 653). Analysis was adjusted for age, sex, miRNA batch, the first three genetic principal components, hematocrit, platelet count, smoking, education, and BMI. Subjects with low MCS and additionally carrying APOE ε4 revealed higher Ct-values for hsa-miR-107, meaning that the level of hsa-miR-107 was reduced in these subjects compared to their genetic counterparts.

**Table 1 biomedicines-10-01560-t001:** Sample characteristics of TREND and LEGEND (consensus sample; for TREND, complete cases for adjustment set 2 and APOE ε4 status; for LEGEND, complete cases for adjustment set 1 and APOE ε4 status). For metric variables, mean, standard deviation, and range are given; for categorical variables, counts and percentages are given.

Variable	TREND (*n* = 3964)	LEGEND (*n* = 2322)	Comparison
**Age**	51.9 (15.4), [20–83]	55.9 (14.3), [29–89]	T = −9.06, *p* < 2.2 × 10^−16^
**Sex**			Chi2 = 0.45, *p* = 0.50
**Males**	1919 (48.4%)	1103 (47.5%)	
**Females**	2045 (51.6%)	1219 (52.5%)	
**Education**			Chi2 = 45.07, *p* = 1.64 × 10^−10^
**<10 years**	909 (22.9%)	677 (29.2%)	
**=10 years**	2052 (51.8%)	1196 (51.5%)	
**>10 years**	1003 (25.3%)	449 (19.3%)	
**APOE ε4 status**			Chi2 = 3.55, *p* = 0.06
**ε4 carrier**	951 (24%)	508 (21.9%)	
**non carrier**	3013 (76%)	1814 (78.1%)	
**MDD**			Chi2 = 2.38, *p* = 0.12
**No**	3153 (81.5%)	1882 (83.1%)	
**Yes**	718 (18.5%)	384 (16.9%)	
**NA**	93	56	
**Mental health score (MCS)**	52.5 (8.6), [16.6–70.3]	51.9 (9.3), [15.6–70.8]	T = 2.72, *p* = 0.007
**NA**	93	0	
**Depressive symptoms (PHQ–9)**	3.9 (3.6),[0–26]	-	
**NA**	208		
**Depressive symptoms (BDI-II)**	-	6.5 (7.3), [0–54.6]	
**NA**		91	
**NAI**		-	
**Immediate recall**	5.2 (1.4), [0–8]		
**NA**	3		
**NAI**		-	
**Delayed recall**	5.7 (1.7), [−3–8]		
**NA**	28		
**VLMT**	-		
**Immediate recall**		24.4 (6.4), [1–45]	
**NA**		142	
**VLMT**	-		
**Delayed recall**		7.9 (3.1), [0–15]	
**NA**		236	
**Smoking status**			Chi2 = 32.26, *p* = 9.9 × 10^∓8^
**Never**	1444 (36.4%)	720 (38.3%)	
**Former**	1461 (36.9%)	783 (41.7%)	
**Current**	1059 (26.7%)	376 (20.0%)	
**Diabetes**			Chi2 = 10.74, *p* = 0.001
**No**		1968 (85.1%)	
**Yes**	3490 (88%)	344 (14.9%)	
**NA**	474 (12%)	10	
**Alcohol (beverage last 12 month)**		-	
**Never**	430 (10.8%)		
**once per month**	1093 (27.6%)		
**2–4 per month**	1247 (31.5%)		
**2–3 per week**	791 (20.0%)		
**>3 per week**	403 (10.8%)		
**Hypertension**			Chi2 = 34.74, *p* = 3.8 × 10^−9^
**No**		1046 (55.9%)	
**Yes**	2076 (52.4%)	824 (44.1%)	
**NA**	1888 (47.6%)	452	
**Heart disease**			Chi2 = 1026, *p* < 2.2 × 10^−16^
**No**		929 (45.4%)	
**Yes**	3363 (84.8%)	1117 (54.6%)	
**NA**	601 (15.2%)	443	
**Time between SHIP-START-2 and LEGEND in years**		1.6 (1.2), [−1.8–4.9]	

VLMT: Verbal Learning and Memory Test; NAI: Nuremberg Age Inventory; PHQ-9: Patient Health Questionnaire; BDI-II: Beck Depression Inventory; MDD: Major Depressive Disorder; MCS: Mental Health Composite Score of the SF-12; NA: not available/not answered; -: questionnaire/variable not part of examination.

**Table 2 biomedicines-10-01560-t002:** Direct effects of depression variables and APOE ε4 status on the endpoint verbal memory (NAI) in TREND under the adjustment of two different covariate sets.

Adjustment	Predictor	Immediate Recall n, β, *p*-Value	Delayed Recall n, β, *p*-Value
**Covariate set 1**	**PHQ-9**	**3753, −0.0198, 0.00026**	**3731, −0.022, 0.0027**
	**MCS**	**3869, 0.008, 0.00024**	**3845, 0.008, 0.0088**
	**MDD**	3868, −0.012, 0.812	3844, −0.052, 0.443
	**APOE** ε4	**3964, −0.08, 0.073**	**3936, −0.10, 0.097**
**Covariate set 2**	**PHQ-9**	**3753, −0.018, 0.0127**	**3731, −0.021, 0.005**
	**MCS**	**3869, 0.0073, 0.0023**	**3845, 0.007, 0.017**
	**MDD**	3868, 0.00012, 0.998	3844, −0.052, 0.436
	**APOE ε4**	**3964, −0.083, 0.0596**	**3936, −0.1, 0.094**

PHQ-9: Patient Health Questionnaire; MCS: Mental Health Composite Score of the SF-12; MDD: Major Depressive Disorder; covariate set 1: age, sex, and education; covariate set 2: age, sex, education, hypertension, smoking status, alcohol intake, diabetes, and heart disease; for APOE ε4, additional adjustment for the first 3 genetic principal components and genetic batch; bold: significant results for one-sided significance level *p* < 0.1.

**Table 3 biomedicines-10-01560-t003:** Results of the interaction analyses between depression variables and APOE ε4 status on the endpoint verbal memory (NAI) in TREND under the adjustment of two different covariate sets.

Adjustment	Interaction Term	Immediate Recall n, β, *p*-Value	Delayed Recall n, β, *p*-Value
**Covariate set 1**	**PHQ-9 × APOE ε4**	3753, −0.019, 0.106	3731, −0.006, 0.72
	**MCS × APOE ε4**	**3869, 0.009, 0.097**	3845, −0.003, 0.704
	**MDD × APOE ε4**	3868, 0.013, 0.91	3844, −0.033, 0.83
**Covariate set 2**	**PHQ-9 × APOE ε4**	**3753, −0.021, 0.083**	3731, −0.008, 0.633
	**MCS × APOE ε4**	**3869, 0.009, 0.087**	3845, −0.002, 0.77
	**MDD × APOE ε4**	3868, 0.007, 0.951	3844, −0042, 0.78

NAI: Nuremberg Age Inventory; PHQ-9: Patient Health Questionnaire; MCS: Mental Health Composite Score of the SF-12; MDD: Major Depressive Disorder; covariate set 1: age, sex, and education; covariate set 2: age, sex, education, hypertension, smoking status, alcohol intake, diabetes, and heart disease; for APOE ε4, additional adjustment for the first 3 genetic principal components and genetic batch; bold: significant results for one-sided significance level *p* < 0.1.

**Table 4 biomedicines-10-01560-t004:** Direct effects and interaction results for depression variables and APOE4 status on the endpoint verbal memory (VLMT) in LEGEND for covariate set 1.

Predictor	Immediate Recall n, β, *p*-Value	Delayed Recall n, β, *p*-Value
**BDI-II**	**2132, −0.06, 0.00016**	**2046, −0.025, 0.0045**
**MCS**	**2180, 0.032, 0.01**	**2086, 0.016, 0.017**
**MDD**	2162, 0.244, 0.42	2078, −0.06, 0.73
**APOE ε4**	2180, 0.23, 0.41	2086, 0.066, 0.677
**Interaction term**		
**BDI × APOE ε4**	**2132, −0.066, 0.075**	2046, −0.027, 0.167
**MCS × APOE ε4**	2180, 0.0041, 0.88	2086, −0.0045, 0.76
**MDD × APOE ε4**	2162, −1.22, 0.12	2078, −0.53, 0.21

VLMT: Verbal Learning and Memory Test; BDI-II: Beck Depression Inventory; MCS: Mental Health Composite Score of the SF-12; MDD: Major Depressive Disorder; adjusted for age, sex, and education; for APOE ε4 additional adjustment for the first 3 genetic principal components; bold: significant results for one-sided significance level *p* < 0.1.

**Table 5 biomedicines-10-01560-t005:** Direct effects and interaction results for depression variables and APOE4 status on the endpoint verbal memory (VLMT) in LEGEND for covariate set 2.

Predictor	Immediate Recall n, β, *p*-Value	Delayed Recall n, β, *p*-Value
**BDI-II**	**1755, −0.04, 0.019**	**1685, −0.021, 0.041**
**MCS**	**1778, 0.025, 0.078**	**1704, 0.015, 0.047**
**MDD**	1767, 0.36, 0.31	1697, −0.19, 0.32
**APOE ε4**	1778, 0.11, 0.72	1704, −0.034, 0.83
**Interaction term**		
**BDI × APOE ε4**	1755, −0.057, 0.172	1685, −0.013, 0.56
**MCS × APOE ε4**	1778, 0.005, 0.86	1704, 0.0003, 0.99
**MDD × APOE ε4**	1767, −1.16, 0.17	1697, −0.64, 0.18

VLMT: Verbal Learning and Memory Test; BDI-II: Beck Depression Inventory; MCS: Mental Health Composite Score of the SF-12; MDD: Major Depressive Disorder; adjusted for age, sex, education, hypertension, smoking status, diabetes and heart disease, and time between SHIP-START-2 and LEGEND; for APOE ε4, additional adjustment for the first 3 genetic principal components; bold: significant results for one-sided significance level *p* < 0.1.

**Table 6 biomedicines-10-01560-t006:** Results of the interaction analyses (APOE ε4*depression) for all miRNAs with a *p*-value < 0.01.

Micro RNA	APOE ε4*PHQ-9 β, *p*-Value	APOE ε4*MCS β, *p*-Value
** *hsa* ** **-miR-221-3p**	−0.094, 0.002	0.027, 0.029
** *hsa* ** **-miR-376a-3p**	−0.094, 0.005	0.036, 0.012
** *hsa* ** **-let-7d-3p**	−0.042, 0.004	0.015, 0.018
** *hsa* ** **-miR-107**	0.024, 0.016	**−0.027, 0.0001**
** *hsa* ** **-miR-382-5p**	−0.084,0.011	0.043, 0.002
** *hsa* ** **-miR-181a-5p**	0.041, 0.069	−0.028, 0.005
** *hsa* ** **-miR-99a-5p**	0.036, 0.098	−0.026, 0.007
** *hsa* ** **-miR-222-3p**	0.006, 0.678	−0.017, 0.009

PHQ-9: Patients Health Questionnaire; MCS: Mental Health Composite Score of the SF-12; bold: significant after multiple testing correction.

## Data Availability

Research data are available after formal application to the SHIP review board (https://www.fvcm.med.uni-greifswald.de/cm_antrag/index.php, accessed on 20 May 2022).

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
