# Peer review of "APOE ε4 in Depression-Associated Memory Impairment—Evidence from Genetic and MicroRNA Analyses"

_biomedicines, 2022, doi:10.3390/biomedicines10071560_

Round 1

Reviewer 1 Report

The manuscript by Bonk and co-workers intended to elucidate the close connection between depression-related cognitive deficits and Alzheimer disease (AD). In particular, the authors focused in investigating if the presence of the APOE ε4 status conditioned such association. Additional, the authors investigated the potential role of circulating plasma miRNAs as potential epigenetic regulators of depression-AD association.

Overall, this is an interesting and timely work as the prevalence of both diseases is exponentially increasing worldwide. The manuscript is nicely written and structured and results certainly add significant value to the field and will reach a vast audience.

Author Response

The manuscript by Bonk and co-workers intended to elucidate the close connection between depression-related cognitive deficits and Alzheimer disease (AD). In particular, the authors focused in investigating if the presence of the APOE ε4 status conditioned such association. Additional, the authors investigated the potential role of circulating plasma miRNAs as potential epigenetic regulators of depression-AD association.

Overall, this is an interesting and timely work as the prevalence of both diseases is exponentially increasing worldwide. The manuscript is nicely written and structured and results certainly add significant value to the field and will reach a vast audience.

Answer: We want to thank reviewer 1 for this positive feedback.

Reviewer 2 Report

In this manuscript, Bonk and colleagues describe their work on identifying a link between ApoE4 and depression on memory functions in two communal cohorts of non-demented subjects. The study further attempts to find a plausible biological link between ApoE4, depression, memory functions, and circulating miRNA. The study is interesting and has merit; however, I have some reservations regarding the technical soundness of the study that diminishes my enthusiasm for it. My comments are listed below in the order of perusing the manuscript. 

1. It is unclear whether SHIP-START-2 and LEGEND are a part of the same cohort of subjects? If yes, there may be a reasonable justification for using the covariates information from SHIP-START-2; however, if not, then these covariates cannot be really used as a substitute across the 2 samples.  While I applaud the authors to attempt to correct for the confounding effects of these variables, I consider "transplanting" the covariates from the SHIP-START-2 sample onto LEGEND, questionable if not detrimental to the study. 

2. How was the genetic data harmonization between the different platforms done to ensure that consistency across the two platforms. 

3. Is the imputation quality >0.8, the posterior probability? How was the value of 0.8 determined?  

4. The authors say the miRNA data were available on a subset of the TREND sample. On how many individuals were these miRNA data available? ? What did their demographic data look like? Also, state here how many miRNAs were assessed in total and what miRBase version are they derived from. In the suppl. material only 179 miRNAs were assessed. How were these selected? If I understood correctly, these were selected by the manufacturer of the RT-PCR miRNA plate; however, based on what criteria? Since this is a PCR based detection, additional details such as detection biochemistry and cycling conditions need to be specified. Also, how was the PCR efficiency determined to ensure the accuracy of the PCR results across the different miRNA? Additional information regarding the approach used to randomize the samples across the two batches is also needed. 

5. The authors say miRNAs present at least in 40% of the sample were retained for further analyses. Is that percentage for the entire sample, or for each batch? 

6. at what step were the spike-ins introduced in the experiment? I assume before the miRNA isolation from blood? 

7. The analysis using two rounds of covariates adjustment is not clearly described. What results from the first adjustment were used in the second adjustment? Why were not all covariates used in a single model? 

8. Table 3. is confusing. In the text, the authors state "...significant interaction effect between current depressive symptoms and 269 APOE ε4 status on immediate recall, but not on delayed recall of words." What I do not understand is why the authors did not report the significance of the interaction term between PHQ-9*APO E4 but are reporting the significance between the interaction term and covariates set 1 and 2? Even so, non of these covariates seem to be significant! Am I missing something here? 

Reviewer 3 Report

First of all, I would like to congratulate the authors for their work and for having chosen to study the cognitive impact of depression and the association with the APOE E4 allele. 

Next, I would like to point out some aspects that need to be improved and that raise certain doubts in order to consider the acceptance of the work in the present form: 

1) The multidomain cognitive impairment described in depression and other affective disorders is usually multidomain with a disejective predominance, why has a more complete neuropsychological evaluation not been performed? 

2) Having an APOE E4 allele is the major risk factor for sporadic Alzheimer's disease and even in genetically determined forms may have an impact. The APOE E4 allele has been associated as a risk factor for other conditions such as epilepsy, a little more information should be added in this regard. 

3) No biomarkers are available to rule out the existence of Alzheimer's disease (AD) copathology in the subjects included in the study. They could well have prodromal Alzheimer's disease and depressive symptoms in that context.

I believe that in this sense the design of the study does not allow a correct assessment of the impact of APOE E4 in the context of isolated depression on the memory domain. 

It would make more sense to assess whether in those with biomarker-supported prodromal AD having APOE E4 carries a higher risk of depression and/or severity of depression and in those subjects with core biomarkers of negative AD to assess whether depression is more prevalent in the APOE E4 group. 

4) Although epigenetic biomarkers are promising, to date their use has not been approved due to difficulties in the performance of results between groups, lack of specificity,... in the context of Alzheimer's disease. These limitations should be further emphasized. To assess whether epigenetic biomarkers allow us to better understand the pathogenesis of diseases and thus help to identify possible new therapeutic targets. 

I will not continue with the review until I know the response that the authors provide to what has already been pointed out.

Author Response

Reviewer 3

First of all, I would like to congratulate the authors for their work and for having chosen to study the cognitive impact of depression and the association with the APOE E4 allele.

Next, I would like to point out some aspects that need to be improved and that raise certain doubts in order to consider the acceptance of the work in the present form:

1) The multidomain cognitive impairment described in depression and other affective disorders is usually multidomain with a disejective predominance, why has a more complete neuropsychological evaluation not been performed?

Answer: SHIP is a general population study with a lot of different medical and non-medical aspects that have been collected, also outside neuropsychiatric domains and conditions. However, in such a large study, there is not much room for a detailed and comprehensive assessment of single domains, such as different aspects of cognition. We have also included this in our limitation section. In our manuscript we are also focusing on verbal memory as this ist he only cognitive domain we have measured. More comprehensive data on cognitive domains in SHIP are simply not available. Nevertheless, we try to transfer our results to cognition in general in the discussion.

2) Having an APOE E4 allele is the major risk factor for sporadic Alzheimer's disease and even in genetically determined forms may have an impact. The APOE E4 allele has been associated as a risk factor for other conditions such as epilepsy, a little more information should be added in this regard.

Answer: We included studies on the association between APOE e4 allele and other neuropsychiatric disorders in the introduction part, such as epilepsy. As such data is not available in SHIP, we are not able to correct for these conditions in our analyses.

Thus, we added the following sentence: „Associations have also been reported for other neuropsychiatric conditions, such as epilepsy, MDD or schizophrenia.“

Altuna, M.; Olmedo-Saura, G.; Carmona-Iragui, M.; Fortea, J. Mechanisms Involved in Epileptogenesis in Alzheimer’s Disease and Their Therapeutic Implications. Int J Mol Sci. 2022, 23, 4307, doi:10.3390/ijms23084307.

Forero, D.A.; Lopez-Leon, S.; Gonzales-Giraldo, Y.; Dries, D.R.; Pereira-Morales, A.J.; Jimenez, K.M.; Franco-Restrepo, J.E. APOE gene and neuropsychiatric disorders and endophenotypes: A comprehensive review. Am J Med Genet B Neuropsychiatr Genet. 2018. 177. 126-142. Doi:10.1002/ajmg.b.32516.

3) No biomarkers are available to rule out the existence of Alzheimer's disease (AD) copathology in the subjects included in the study. They could well have prodromal Alzheimer's disease and depressive symptoms in that context.

Answer: You are right that we had no information on AD or biomarkers of AD in our study. However, as this was a general population study, the proportion of AD cases might be very low and not have an impact on our study results. To acknowledge this point, we included it into the study limitations. In SHIP-TREND-0 we also have information on the current intake of antidementiva medication. But either adjusting the analysis for this medication nor excluding these individuals (n=30) changed the significance of our results.

I believe that in this sense the design of the study does not allow a correct assessment of the impact of APOE E4 in the context of isolated depression on the memory domain.

It would make more sense to assess whether in those with biomarker-supported prodromal AD having APOE E4 carries a higher risk of depression and/or severity of depression and in those subjects with core biomarkers of negative AD to assess whether depression is more prevalent in the APOE E4 group.

Answer: We do not believe that in our setting we could talk about „biomarker-supported prodromal AD“. We just do not have the diagnostic or the biomarker information to selects prodromal AD with high accuracy. This would only be speculative and not further inform or strengthen our results. Throughout our paper we are talking about subjects with better/worse memory performance. In our current setting, we are only able to identify a statistical association between APOE e4 allele and cognition and a moderative effect of the presence of current depressive symptoms. We are trying to put this into a possible biologic mechanistic concept. Further research with more AD specific data (humans or animal studies) are needed to support our conclusions.

4) Although epigenetic biomarkers are promising, to date their use has not been approved due to difficulties in the performance of results between groups, lack of specificity,... in the context of Alzheimer's disease. These limitations should be further emphasized. To assess whether epigenetic biomarkers allow us to better understand the pathogenesis of diseases and thus help to identify possible new therapeutic targets.

Answer: You are right that one limitation of miRNAs as biomarkers for complex diseases is the poor replication of results between studies. Nevertheless, we are not claiming that our identified miRNA is a marker for AD. We were able to identifiy a changed miRNA level in subjects with current depressive symptoms and additionally carrying the APOE e4 allele and put this into a biological context. We were able to refer to other studies that identified an association of this miRNA with AD and also could link this miRNA to amyloid beta in the biological context.

I will not continue with the review until I know the response that the authors provide to what has already been pointed out.

Answer: We hope that our answers and explanations might be sufficient to rule out misunderstandings. Again, we want to point out that our study is neither a clinical study or a biomarker search. It is an epidemiological study revealing significant statistical associations that can be put into a biological framework. Additional verification and elucidation of the mechanisms must be undertaken using more appropriate data.

Reviewer 4 Report

In the manuscript by Bonk et al., the authors investigated the impact of the APOE ε4 allele on verbal memory and the interaction with depression in two large general population cohorts from the Study of Health in Pomerania. They confirmed both depression and APOE ε4 allele was associated with impaired memory performance. Further miRNA data revealed reduced levels of hsa-miR-25 107 in subjects with current depressive symptoms and carrying APOE e4. The work is of importance to the field. However, there are still several concerns that need to be addressed. Please see below for specific comments: 

1. In the manuscript, the authors claimed they used LEGEND cohort as a replication to the results of TREND cohort. However, some of the evaluation methods between these studies were different, and not all findings from TREND can be replicated in TREND . The authors should discuss more about how different evaluation methods potentially influence the results. 

2. APOE4 female has been reported to show more severe cognition impairment in AD patients. Is there any gender effects of APOE4 in depression and verbal memory?

3. All of the miRNA analysis results were based on the omics data without no validation. The authors should try to validate some important miRNA level changes, like hsa-miR-107, using human brain samples

Author Response

In the manuscript by Bonk et al., the authors investigated the impact of the APOE ε4 allele on verbal memory and the interaction with depression in two large general population cohorts from the Study of Health in Pomerania. They confirmed both depression and APOE ε4 allele was associated with impaired memory performance. Further miRNA data revealed reduced levels of hsa-miR-25 107 in subjects with current depressive symptoms and carrying APOE e4. The work is of importance to the field. However, there are still several concerns that need to be addressed. Please see below for specific comments: 

  1. In the manuscript, the authors claimed they used LEGEND cohort as a replication to the results of TREND cohort. However, some of the evaluation methods between these studies were different, and not all findings from TREND can be replicated in TREND. The authors should discuss more about how different evaluation methods potentially influence the results. 

Answer: Thank you for this critical comment. Although, both samples are taken from the same underlying population, you are right that in some measurements both cohorts are different, e.g. current depressive symptoms (PHQ-9 vs. BDI-II) or verbal memory (NAI vs. VLMT). Although, both measure the same underlying construct, the measurements are different. This can either be seen as a limitation as this is no direct replication or as a strength of results as replication is possible under different methodological conditions. Nevertheless, you are right that this should be pointed out more in detail in the discussion. We therefore added the following statement:

“Both samples were collected from the same study population in northeast Germany. Unfortunately, questionnaires to measure current depressive symptoms and verbal memory were not identical. This can either be viewed as a limitation or as a strength of the robustness of the results as the significant effect was generalizable.”

  1. APOE4 female has been reported to show more severe cognition impairment in AD patients. Is there any gender effects of APOE4 in depression and verbal memory?

Answer: We thank reviewer 4 for this interesting biological question. In sex stratified analyses no significant differences could be observed. Directions of effects were identical for males and females, although effect sizes were different in some cases. As the individual samples were smaller effects did not remain significant due to power issues.

  1. All of the miRNA analysis results were based on the omics data without no validation. The authors should try to validate some important miRNA level changes, like hsa-miR-107, using human brain samples

Answer: You are correct that we were not able to replicate the miRNA findings as we had no replication sample available. The miRNA results did not serve as our main findings. They were used to support our results on the interaction between depressive symptoms and APOE e4 status and results revealed a link towards AD. As our data were measured in plasma, we also tried to draw a link towards brain miRNA using data from the human miRNA tissue atlas. Unfortunately, human brain samples were not available in our study.

Round 2

Reviewer 2 Report

I have no further comments for the authors. 

Reviewer 3 Report

First of all, I would like to thank the authors for their efforts to incorporate most of the suggestions made. I believe that this has significantly improved the quality of the work. 

Although it is true that the purpose of the work is not to correlate with AD biomarkers, I believe that the work would have gained robustness and significance if AD biomarkers and proper cognitive evaluation has been performed.  I think it is wrong to point out that prodromal Alzheimer's disease would be very rare because in other population studies it is quite frequent (it is underestimated at the population level) and I think that the incorporation of cognitive screening tests would have been necessary to add robustness (and it is done in epidemiological studies. There are population based cohorts with hundreds of participants with cognitive screening incorporating the assessment of affective and conductual symptoms). I leave this as a consideration for future work by the research team, since I am aware that at the present time this is an aspect that cannot be remedied.

are

I have no further comments to add.

Reviewer 4 Report

The authors have addressed all of my comments and concerns. I have no additional comments.